# Nutrient enrichment induces dormancy and decreases diversity of active bacteria in salt marsh sediments

Patrick J. Kearns[1], John H. Angell[1], Evan M. Howard[2], Linda A. Deegan[3,†], Rachel H.R. Stanley[4] & Jennifer L. Bowen[1,†]

Microorganisms control key biogeochemical pathways, thus changes in microbial diversity, community structure and activity can affect ecosystem response to environmental drivers. Understanding factors that control the proportion of active microbes in the environment and how they vary when perturbed is critical to anticipating ecosystem response to global change. Increasing supplies of anthropogenic nitrogen to ecosystems globally makes it imperative that we understand how nutrient supply alters active microbial communities. Here we show that nitrogen additions to salt marshes cause a shift in the active microbial community despite no change in the total community. The active community shift causes the proportion of dormant microbial taxa to double, from 45 to 90%, and induces diversity loss in the active portion of the community. Our results suggest that perturbations to salt marshes can drastically alter active microbial communities, however these communities may remain resilient by protecting total diversity through increased dormancy.

[1] Department of Biology, University of Massachusetts Boston, 100 Morrissey Boulevard, Boston, Massachusetts 02125, USA. [2] Department of Chemistry and Geochemistry, Woods Hole Oceanographic Institution (MIT-WHOI Joint Program), 266 Woods Hole Road, Woods Hole, Massachusetts 02543, USA. [3] Ecosystems Center, Marine Biological Laboratory, 7 MBL Street, Woods Hole, Massachusetts 02543, USA. [4] Department of Chemistry, Wellesley College, 106 Central Street, Wellesley, Massachusetts 02481, USA. † Present addresses: Woods Hole Research Center, 149 Falmouth Road, Falmouth, Massachusetts 02540, USA (L.A.D.); Department of Marine and Environmental Science, Marine Science Center, Northeastern University, 430 Nahant Road, Nahant, Massachusetts 01908, USA (J.L.B.). Correspondence and requests for materials should be addressed to J.L.B. (email: je.bowen@northeastern.edu).

Human activities have increased the amount of reactive nitrogen (N) in the biosphere. Excess bioavailable N entering ecosystems can elicit many deleterious effects[1], including decreased biodiversity[2,3]. Numerous studies have indicated the important coupling between biodiversity and ecosystem function[4], though the emphasis is typically on macro-organismal diversity. Microorganisms, however, exert strong controls over ecosystems through the mediation of key biogeochemical cycles, including the N cycle. Despite their importance to ecosystem function, the relationship between microbial community composition, diversity and ecosystem function is not clearly elucidated[5]. Resolving the responses of microbial communities, in particular the active taxa that maintain ecosystem function, is essential for predicting how ecosystems will respond to global changes such as increased anthropogenic N supply.

Analysis of the 16S rRNA gene, which is commonly used to assess bacterial community structure, accounts for all cells, including cells that are active, dormant and recently dead, as well as extracellular DNA[6]. Relic DNA from dead cells has been shown to mask ecologically important patterns in soil environments[7]. A large number of dormant or inactive cells might also mask shifts in the active microorganisms that are responsible for critical ecosystem services. Dormancy has been described as a bet-hedging strategy where microbes enter a low metabolic or inactive state when they encounter unfavourable environmental conditions[8]. Microbes can persist in this state until environmental conditions favour their successful growth. Molecular analyses suggest that dormant microbial taxa can account for ∼20–50% of the microbial community depending on the ecosystem and the heterogeneity of the environment[8]. It is unclear how high proportions of dormant taxa affect microbial function, however, dormancy acts as a genomic reservoir allowing for the preservation of genetic diversity in the presence of unfavourable conditions, and may therefore provide an important long-term strategy for maintaining ecosystem function in the face of environmental perturbations[8].

One such perturbation, coastal nutrient supply, especially by nitrogen (N) in the form of nitrate ($NO_3^-$), has resulted in salt marshes being one of the most nutrient-enriched ecosystems in the world, with some systems experiencing nutrient enrichment >500 kg N per km per year[9]. Salt marsh area has significantly declined across the globe and recent evidence suggests that excess nutrients accelerate marsh collapse[10]. Nutrient-induced loss of marsh area is hypothesized to occur through a decrease in belowground plant biomass and an increase in belowground bacterial respiration, resulting in decreased stability of the marsh edge and loss of marsh area. However, evidence from 16S rRNA gene analysis suggests that N-enrichment has a minimal effect on salt marsh sediment total bacterial community structure[11]. The lack of response in the total bacterial community is surprising given the observed shifts in biogeochemical function and respiration as a result of nutrient enrichment[12,13] and additional work that clearly connects changes in ecosystem function with changes in bacterial community structure[14].

We hypothesize that the apparent lack of bacterial community response to N-enrichment is a result of a high degree of bacterial dormancy in salt marsh sediments and that the active community of bacteria will shift in response to increased N supply. Salt marsh sediments are highly dynamic habitats, where microorganisms are exposed to variable light, oxygen, salinity, water, carbon and nutrients that can change in minutes to hours. Because of these widely changing conditions, we predict that dormant taxa may account for a substantial portion of the microbial community in marsh sediments. We hypothesize that excess N will favour a small number of taxa that are able to respond to increased N availability and ultimately result in an increase in the portion of dormant cells.

Here we test our hypotheses by using analysis of the 16S rRNA gene and the gene product, 16S rRNA, to assess the response of the total and active microbial communities, respectively, to N-enrichment. Further, we quantified the extent of bacterial dormancy in salt marsh sediments and the effect of nutrient enrichment on active bacterial diversity. Although the relative abundance of 16S rRNA cannot be used as an exact proxy for bacterial growth[15] it does indicate cells that have the potential for growth and cells that are metabolically active, though perhaps not dividing[16]. We examined the effect of experimental long-term nutrient enrichment[10,13,17] on these communities in low and high marsh habitats in a New England salt marsh located at the Plum Island Ecosystems Long-Term Ecological Research Site. Our long-term experimental marsh has been enriched with 70 µM nitrate on the incoming tides seasonally since 2004. Another marsh that we sampled was enriched for 1 year in 2005 and again from 2009 to 2015. In each year enriched marshes received 15-fold more N than reference marshes[13,17].

New England salt marshes are tidal grasslands characterized by low and high marsh habitats that receive different amounts of tidal flooding. Low marsh is dominated by the tall ecotype of *Spartina alterniflora* and is flooded on every semi-diurnal tide. The low marsh habitat receives considerably more N delivered by tidal water than the high marsh, which is dominated by *Spartina patens* and is flooded by only 30% of high tides[17]. We collected triplicate sediment samples from both habitats monthly in each marsh (May–October; 2005, 2006, 2013 and 2014) to examine short-term and long-term trajectories in community structure and activity. After extracting DNA and RNA, we amplified and sequenced the 16S rRNA gene and 16S rRNA to describe the total and active bacterial communities. To examine how changes in the active microbial community might translate to ecosystem scale processes we also measured whole ecosystem metabolism by examining dissolved oxygen concentrations in creek water of one enriched and one reference creek. Our results indicate that nutrient enrichment increases the proportion of dormant cells in salt marsh sediments and markedly decreases the diversity of active bacterial taxa despite no shifts in the total community as a result of this environmental perturbation. These results suggest that salt marsh microbial communities are resilient to perturbation through the maintenance of a robust community of dormant taxa that can respond to environmental change.

## Results

**Total and active community composition.** Our results revealed a sharp division between the active and total microbial communities, suggesting the structure of the active community does not reflect the total community (Fig. 1a, permutational multivariate analysis of variance (PERMANOVA)[18]; $F = 19.43$, $P < 0.001$). There was no significant effect of N-enrichment on total community structure (Fig. 1b; $P > 0.67$) or diversity (Supplementary Fig. 1; $P > 0.43$, although we found a significant effect of habitat on total bacterial community structure ($F = 19.51$, $P < 0.001$). These results extend previous findings that the total salt marsh bacterial community is resistant to perturbation by nutrient enrichment[11] and indicate that this resistance has persisted despite a decade of N-enrichment. In addition, our results demonstrate a surprisingly low amount of variability in the total microbial community, suggesting long-term stability of sediment communities despite perturbations, a phenomenon that was also observed in other New England marshes receiving excess nutrients for over 40 years[11].

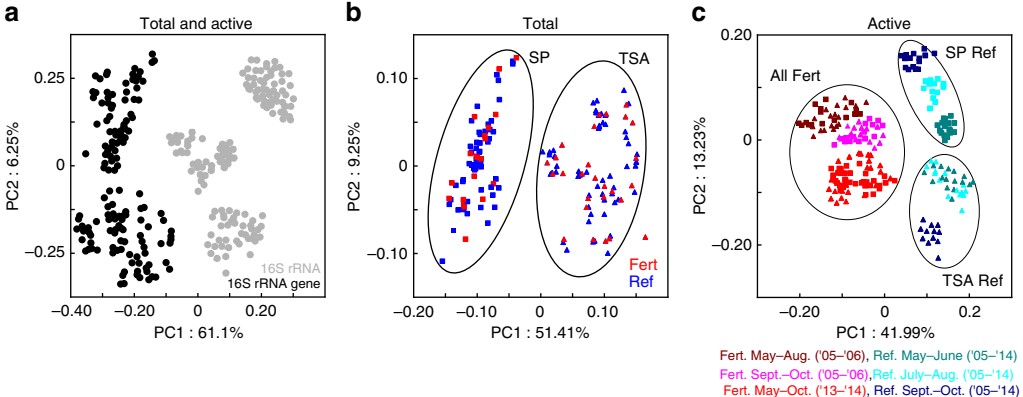

**Figure 1 | Fertilization alters active microbial community composition but not total microbial community composition.** Principal coordinate analysis (PCoA) of weighted UniFrac similarity illustrating significant differences between total and active communities (**a**; PERMANOVA; $F = 19.43$, $P < 0.001$), but no significant difference as a result of fertilization on the total community (**b**) as assessed by the 16S rRNA gene ($P > 0.67$). By contrast there was a significant effect of fertilization on the active community (**c**) assessed by 16S rRNA ($F = 7.75$, $P < 0.01$). These data derive from samples collected within 1 m of a permanent transect line that bisected two marsh habitats in four marsh creeks and were collected monthly from May to October in 2005, 2006, 2013 and 2014 ($N = 192$). Fert, Fertilized; Ref, reference.

Analysis of the potentially active bacterial community shows a markedly different pattern than the total bacterial community and indicates that long-term nutrient enrichment spatiotemporally standardizes the active community, overriding the importance of both habitat and season as structuring forces (Fig. 1c). In reference marshes, habitat (PERMANOVA; $P < 0.01$, $F = 9.05$) and seasonality ($P < 0.03$, $F = 24.24$) structured active bacterial communities and these patterns were repeatable over a decade. Active bacterial communities in N-enriched marshes (Fig. 1c), however, were significantly different from reference marshes and were no longer influenced by habitat-specific factors (Fig. 1c; $P < 0.001$, $F = 7.75$). Further, the seasonal patterns in the active microbial community were initially present in the early years of fertilization (Fig. 1c), however, after a decade of fertilization these patterns were no longer present.

**Changes in dormancy and diversity.** To assess the role of dormancy we calculated the proportion of dormant taxa in each sample using the 16S rRNA:16S rRNA gene ratio and defining any taxon with a ratio $\leq 1$ as dormant[19]. There are caveats to this approach[15], so to ensure our interpretation was robust, we also increased the ratio required from a ratio of one to a ratio of 50. (ref. 20; Supplementary Fig. 2). Regardless of the ratio threshold we used to define dormancy (Supplementary Fig. 2), the results show a consistent pattern that the dormant taxa in N-enriched sediments accounted for a significantly higher proportion of the bacterial community than in reference sediments (Fig. 2a; analysis of variance (ANOVA), $P < 0.001$, $F = 41.94$). Dormant taxa remained $\sim 45\%$ of the community in reference sediments (Fig. 2a) but in N-enriched sediments the proportion of dormant taxa increased over time (Fig. 2a) to $\sim 90\%$ of the community after a decade of N-enrichment.

The increase in dormancy corresponded to a decrease in potentially active bacterial diversity. Reference marsh sediments showed no temporal trends in Shannon Diversity and displayed significantly higher active diversity than N-enriched sediments (Fig. 2b,c; $F = 12.49$, $P < 0.01$). Furthermore, bacterial communities in N-enriched *S. alterniflora* displayed a significant loss in active diversity over time ($P < 0.001$, $F = 12.43$; Fig. 2b). By contrast, although active diversity in *S. patens* appears to decrease over time (Fig. 2c), the variation in diversity was higher and the decrease was not significant. The greater response observed in the

low marsh *S. alterniflora* habitat likely results from the fact that it is inundated for a longer period of time than *S. patens* and therefore receives a greater N supply[17].

**Taxonomic compositional changes.** Owing to the observed sharp decline in active bacterial diversity, we assessed which taxa responded to N-enrichment by examining the ratio of 16S rRNA to the 16S rRNA gene $+ 1$ (Fig. 3). Implicit in this calculation is the assumption that if a sequence is present in the 16S rRNA of a sample it must also be present at least one time on the 16S rRNA gene, but was not detected due to incomplete sequencing. Three of the five most abundant bacterial orders we identified (Desulfobacterales, unclassified Cyanobacteria and Oscillatoriales) had considerably higher ratios in N-enriched compared with reference marshes. Despite having a few taxa with very high 16S rRNA:16S rRNA gene ratios, N-enriched marshes contained fewer abundant active taxa ($n = 23$) than nearby reference marshes ($n = 41$). Furthermore, N-enriched marshes contained a long tail of taxa that had an activity ratio considerably lower than one, indicating that these taxa were largely dormant in N-enriched marshes but remained active in reference marshes. Our results suggest, given the large portion of inactive taxa with very low ratios of 16S rRNA:16S rRNA gene in N-enriched marshes, that many of these taxa are highly abundant members of the total community but are inactive due to N-enrichment enhancing the competitive ability of some taxa at the expense of many others.

Among the most abundant active taxa, defined as those that were present at least 100 times, seven bacterial orders differed significantly (Kruskal–Wallis test, Bonferroni corrected $P < 0.001$) between N-enriched and reference marshes (Fig. 4 and Supplementary Table 1). N-enriched sediments contained large numbers of sequences that were closely related to anaerobic sulfate reducers from the Deltaproteobacterial order Desulfobacterales, as well as to numerous autotrophic Cyanobacteria from the order Oscillatoriales. Whole-genome analysis of *Desulfobacterium autotrophicum* indicates the order Desulfobacterales may be metabolically diverse due to large genomes and numerous transposable elements relative to other sulfate-reducing bacteria[21], which may allow faster response to environmental perturbations, including nutrient enrichment. In reference marshes anoxygenic, or potentially anoxygenic, phototrophs

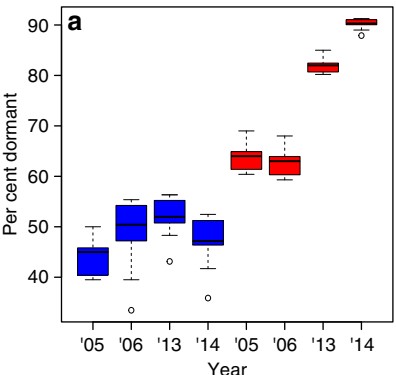
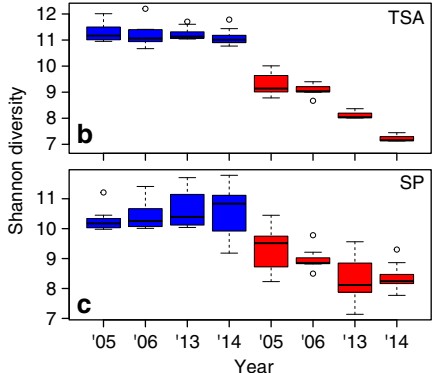

**Figure 2 | Nutrient enrichment increases the per cent of dormant taxa and decreases active diversity.** Per cent of dormant taxa (**a**) in N-enriched (red) and reference (blue) marshes and Shannon diversity of the potentially active community, as assed by 16S rRNA for tall *S. alterniflora* (TSA; **b**) and *S. patens* (SP; **c**). Boxes represent 25–75% quartiles, and the solid black line is the median value. Nutrient enrichment induced a decrease in active diversity ($F = 12.49$, $P < 0.01$, $N = 192$).

(Chromatiales, Chlorobi, Rhizobiales, and Rhodocyclales) were significantly more abundant than in fertilized marshes. By contrast, cyanobacteria dominated the phototrophs in fertilized marshes (Fig. 3), with unclassified Cyanobacteria and the cyanobacterial order Oscillatoriales, responding favourably to nutrient enrichment. Oscillatoriales are common bloom-forming autotrophs that display broad genetic, phenotypic and habitat diversity, which may explain their ability to respond to higher N loads despite many members possessing the capacity to fix N (ref. 22). While N-fixation has been reported in salt marshes[23], which are characterized by high concentrations of particle-bound and pore water ammonium, enhanced N-enrichment to surface sediments may allow Oscillatoria in fertilized marshes to forego N-fixation and increase their activity relative to Oscillatoria in reference marshes.

**Changes in ecosystem metabolism.** To determine whether there were any effects on ecosystem function resulting from the proliferation of Desulfobacterales, unclassified Cyanobacteria and Oscillatoriales, we measured whole ecosystem metabolism using dissolved oxygen concentrations in the water column of one pair of reference and N-enriched marsh tidal creeks (Fig. 5). During high tide, when sediments in the low marsh habitat were submerged, the N-enriched creek demonstrated both enhanced autotrophy during the day and enhanced heterotrophy at night compared with the reference creek (Fig. 5). Previous work also suggested that when salt marsh bacterial activity was enhanced by N-enrichment, it was in response to increased phototrophic productivity[24]. Taken together these data suggest a tight coupling whereby Cyanobacteria in marsh sediments consume excess nitrate and release labile carbon compounds that stimulate the activity of Desulfobacterales. Thus, there may be alteration of the carbon cycle in nutrient-enriched marshes due to enhancement of both bacterial photosynthesis and respiration, and the role these changes play in maintaining marsh geomorphology is a critical concern[10].

**Discussion**
Microbial dormancy in these nutrient-enriched salt marsh sediments is higher than reported values for soil systems[8], lake systems[19] and coastal waters[25] using comparable methodology. These results confirm the assertion that dynamic abiotic conditions may promote dormancy, but also highlight the important role that dormancy plays in the capacity of microbial communities to withstand environmental perturbations.

Dormancy can be a response to nutrient limitation and there is evidence that alleviation of nutrient limitation can reduce rates of dormancy in lake systems[19]. Salt marshes are N-limited systems[26] yet despite N-enrichment, our results indicate that salt marsh microbial communities increase the proportion of dormant taxa in response to N additions, which may suggest a differential response of water and soil/sediment communities. We hypothesize that the high rates of dormancy we observed in the nutrient-enriched marshes are a result of enhanced competition whereby bacteria better adapted to higher N-loads outcompete other bacteria for nutrients and energy sources, leading to an increase in dormancy among other taxa present in the community. In a sense, nutrient addition to marsh sediments may induce blooms of Desulfobacterales and Oscillatoriales, much like estuaries often have algal blooms in response to excess N (ref. 27). Other nutrient enrichment-induced changes to the marsh, including changes in the geomorphology[10] of the marsh creeks and changes in faunal abundances[28] could also influence the proportion of active taxa in marsh sediments. Regardless, our data suggest that despite the marked effect of nutrient enrichment on the active microbial community, the total bacterial community remained unaffected, allowing marshes to maintain a reservoir of genetic diversity that provides a buffer such that the highly diverse bacterial community can respond to future environmental change.

In contrast to terrestrial grasslands where edaphic conditions are comparably more stable, salt marshes are highly dynamic systems, with environmental conditions changing on rapid timescales (minutes to hours). Our results demonstrate a surprising stability of the total microbial community to N-enrichment (Fig. 1b), however, a recent analysis of terrestrial grasslands systems[3] showed consistent shifts in the total microbial community in response to nutrient enrichment from grasslands across the globe. Our results suggest differential control mechanisms on the structure of microbial communities between soils and water-saturated sediments. This difference may be related to community adaptation to intrinsic environmental variability such that sediment microbial communities respond by altering the dormant versus active community, thereby allowing them to respond to rapid changes in their environment, while soil microbial communities alter the total community structure.

Nutrient-enriched salt marshes demonstrate a marked loss in active microbial diversity (Fig. 2b,c), suggesting a net shift in the metabolic functioning of microbial populations, despite no net change in total community composition. Given the stark decline

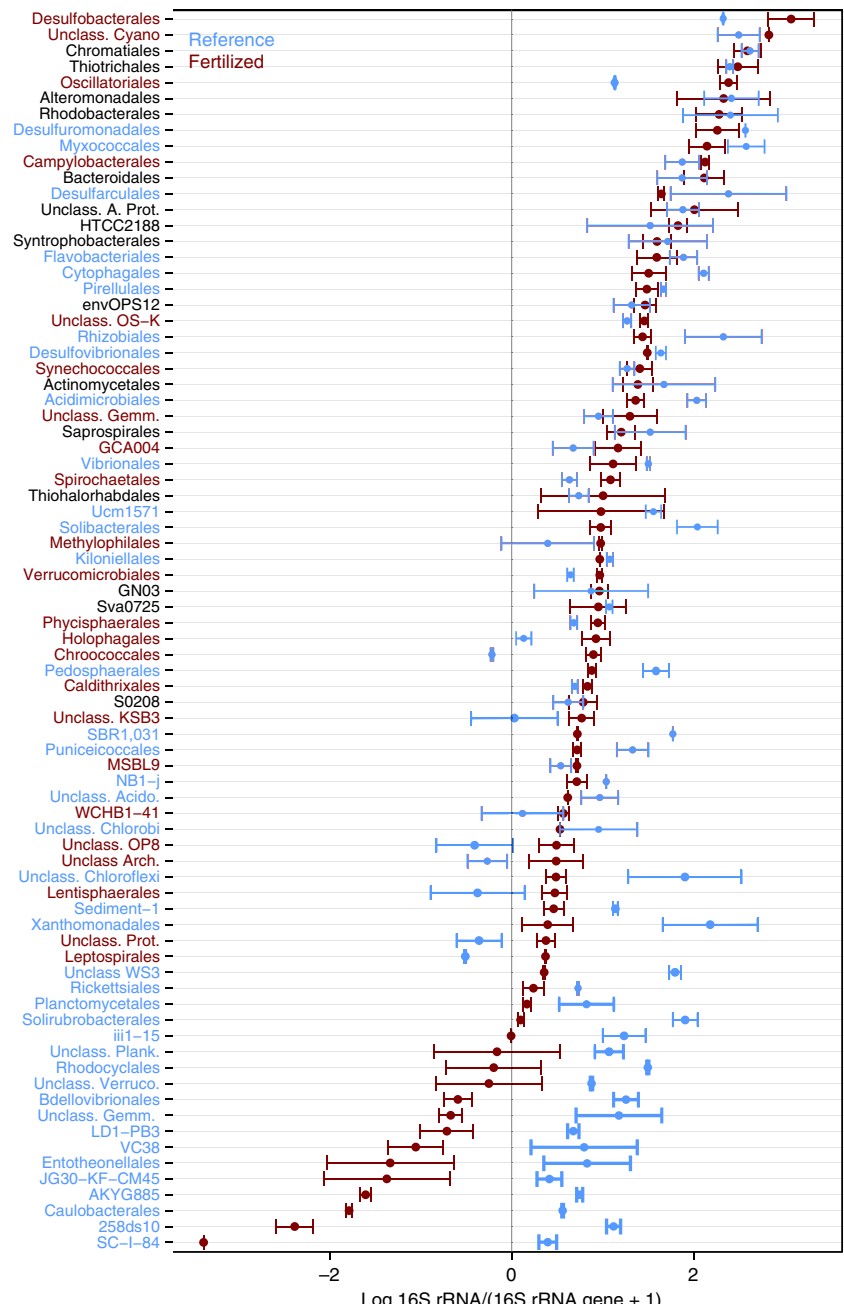

**Figure 3 | Nutrient-enriched marshes have a large number of inactive taxa.** Order-level taxonomic composition of fertilized (red) and reference (blue) sediments assessed by the $\log_{10}$ 16S rRNA:16S rRNA gene ratio for the microbial orders comprising 90% of all sequences. Orders coloured blue are significantly more abundant in reference sediments (Kruskal–Wallis test, Bonferroni corrected $P < 0.001$) while red orders are significantly more abundant in N-enriched sediments. Black line is a 16S rRNA:16S rRNA gene ratio of 1. Taxa with a ratio higher than 1 are considered active and taxa with ratios lower than 1 are considered dormant. Points are the mean and error bars are s.e.m.

in active bacterial diversity we expect to see a decrease in functional diversity, as well as a shift in ecosystem function, as was previously shown in pelagic communities[29,30]. While relationships between biodiversity and ecosystem function have been shown in a variety of organisms such as plants[4], studies examining the relationship between diversity and function in microbial communities have been equivocal. Reed and Martiny[14] demonstrated a net shift in community composition correlated with enhanced respiration, however, this correlation was not observed in a similar study[31]. Given the marked decrease in diversity of the active microbial community and the ecosystem-scale enhancement in both autotrophy and heterotrophy in

response to N-enrichment, we conclude that N-enrichment to salt marshes likely alters metabolic function and thus, ecosystem function of these critically important coastal habitats.

Increased N additions to ecosystems worldwide have fundamentally altered microbial communities and the biogeochemical functions they mediate[4,32]. In coastal salt marshes, N-enrichment promotes an active bacterial community that is spatiotemporally homogenous compared with reference marshes. A decade of nutrient enrichment in marsh sediments resulted in the highest reported proportion of dormant bacterial taxa and induced diversity loss in the active portion of the bacterial community, despite no apparent change in the total bacterial community. We

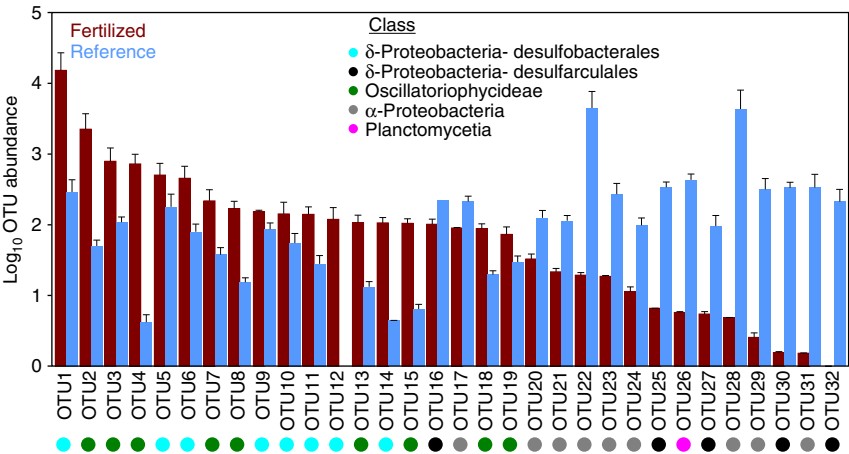

**Figure 4 | Nutrient enrichment increases the abundance of specific taxa.** Significantly different OTUs in the potentially active communities between fertilized (red) and reference (blue) sediments present at least 100 times assessed by a Kruskal–Wallis test (Bonferroni corrected $P < 0.0001$). The class of each OTU is indicated by the coloured circle. More taxonomic information about each OTU can be found in Supplementary Table 1. Points are the mean and error bars are s.e.m.

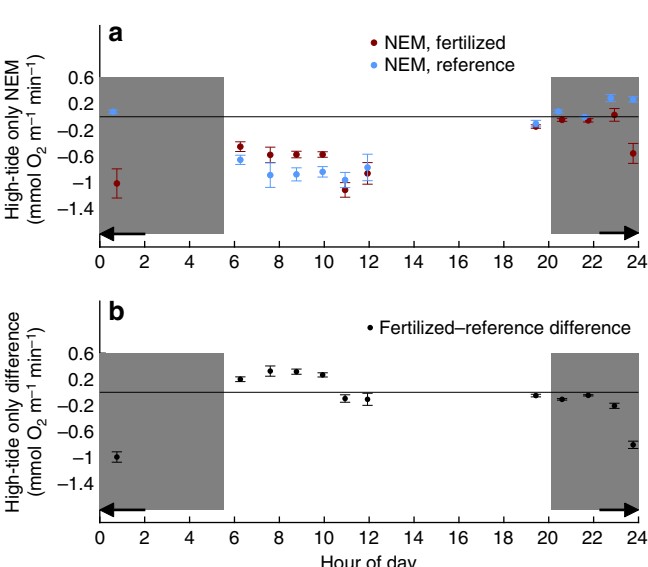

**Figure 5 | Nutrient enrichment also enhances both autotrophy and heterotrophy.** High-tide only dissolved oxygen-based NEM in the fertilized (red) and reference (blue) tidal creeks (**a**) and the difference (black) between the fertilized and reference creeks (**b**) plotted against time of day for each of the 12 high tides during a 6-day period in August 2012. Each point represents the mean NEM or difference rates evaluated every 10 min over the hour of high tide, with one s.d. bar for the means. Note the mean difference rate over each hour of high tide is more tightly constrained than the mean for either creek. Shading denotes the period between dusk and dawn, and black arrows identify the time range of three tides that flood the high marsh platform and may have systematic positive biases in NEM, particularly in the reference creek (see Methods).

suggest that the highly dynamic nature of salt marsh sediments promotes bacterial dormancy as an ecological strategy for the maintenance of microbial diversity. Our results suggest a unique feature of salt marsh bacterial communities in that the active diversity of the community can be markedly diminished, while still maintaining a genetic reservoir of traits that can respond to future environmental changes. This work underscores the importance of examining both total and active bacterial communities in future studies as their responses to global change drivers such as nutrient enrichment are likely to diverge.

## Methods

**Study site description.** This experiment was conducted at the Plum Island Ecosystems Long-Term Ecological Research Station (PIE-LTER) in Northeastern Massachusetts, USA (42.759 N, 70.891 W). Nested within the LTER, a long-term (>10 years), large-scale (60,000 m² per treatment marsh system) nutrient enrichment experiment called the TIDE Project[10,13,17] is ongoing. One marsh system received nutrient enrichment with target enrichment of 70–100 μM nitrate in the creek water on the rising tide continuously throughout the growing season since 2004 and an additional creek was fertilized in 2005 and then again from 2009 to the present. Two additional creeks received no fertilizer and serve as reference marshes.

**Sample collection.** We collected samples monthly over the course of the growing season (May–October) in 2005, 2006, 2013 and 2014 at low tide. All samples were collected within 1 m of permanent transects. Sediment was collected within two dominant macrophytes: *S. patens* and the tall ecotype of *S. alterniflora*. Surface sediments (top 2 cm) were collected in triplicate with sterile, cutoff 30 ml syringes and homogenized in a sterile 50 ml centrifuge tube. Samples were aliquoted into either empty cryovials or cryovials containing 1 ml RNAlater (Invitrogen, Grand Island, NY, USA). Samples put into RNAlater were mixed by vigorous shaking before storage. All samples were stored on dry ice for <1 h until they were flash frozen in liquid nitrogen.

**Nucleic acid extraction.** DNA was extracted from ~0.25 g of sediment using the MoBio PowerSoil Total DNA Isolation Kit (Carlsbad, CA, USA) following the manufacturer's instructions. RNA was extracted from ~0.5 g sediment following the protocol of Mettel *et al.*[33] with modifications. First, to remove residual RNAlater, sediments were spun at 20,000*g* for 1 min and the resulting supernatant was discarded. To each sample 700 μl of PBL buffer (5 mM Tris-HCl (pH 5.0), 5 mM Na₂EDTA, 0.1% (wt/vol) sodium dodecyl sulfate and 6% (vol/vol) water-saturated phenol), along with 0.5 g 0.17 mm glass beads were vortexed at maximum speed for 10 min. Samples were then spun at 20,000*g* for 30 s and the supernatant was transferred to a clean tube. The remaining sediment and glass beads were resuspended in 700 μl TPM buffer (50 mM Tris-HCl (pH 5.0), 1.7% (wt/vol) polyvinylpyrrolidone and 20 mM MgCl₂) and vortexed at maximum speed for an additional 10 min. Sediment was then spun at 20,000*g* for 30 s and the supernatant was pooled with the previous supernatant. An equal volume of phenol:choloroform:isoamyl alcohol (25:24:1 v/v/v) was added to each sample and mixed by vortexing at maximum speed for 30 s. Samples were then spun at 20,000*g* for 30 s, the aqueous layer was transferred to a fresh tube and nucleic acids were precipitated with 0.7 volumes of 100% isopropanol and 0.1 volumes 3 M sodium acetate (pH 5.7). Samples were spun at 20,000*g* for 30 min, the supernatant was discarded and the resulting pellet was washed with 70% ethanol and allowed to air dry. The washed RNA was loaded onto an Illustra Autoseq G-50 Spin Column (GE Healthcare, Pittsburgh, PA, USA), which contained 500 μl prewashed Q-Sepharose (GE Healthcare). Samples were spun at 650*g* for 7 s and then eluted five times with 80 μl of 1.5 M NaCl (pH 5.5). The flow through was transferred to a clean tube, precipitated with 0.7 volumes 100% isopropanol and 0.1 volumes sodium acetate (pH 5.7), and spun at 20,000*g* for 30 min. The resulting pellet was washed with 70% ethanol, allowed to air dry and then resuspended in 50 μl di-ethyl pyrocarbonate-treated water.

RNA samples were checked for DNA contamination using general bacterial primers 515F and 806R (ref. 34), and any DNA contamination was removed using

DNase I (New England BioLabs, Ipswich, MA, USA) following the manufacturer's instructions. We then reverse transcribed RNA to cDNA using the Invitrogen Superscript RT III cDNA synthesis kit following the manufacturer's instructions for random hexamers. Proper cDNA synthesis was verified by PCR with general bacterial primers, with product checked on a 1.5% agarose gel stained with ethidium bromide.

**PCR and sequencing.** We quantified DNA and RNA concentrations with Picogreen and Ribogreen (Invitrogen) kits, respectively, following the manufacturer's instructions, and DNA and cDNA were normalized to $3\,ng\,\mu l^{-1}$ for all PCR reactions. Samples were then prepared for sequencing on the Illumina MiSeq[34]. We first used general bacterial primers 515F and 806R (ref. 34), with appropriate Illumina adaptors, and individual 12 bp GoLay barcodes attached to the reverse primers. We amplified each sample in triplicate using previously described PCR conditions[34]. Samples were verified with gel electrophoresis and target bands were excised and purified with the Qiagen QIAquick gel extraction kit (Qiagen, Valencia, CA, USA). Samples were quantified fluorometrically using a Qubit 2.0 (ThermoFisher, Waltham, MA, USA) and were pooled in equal molar concentrations for sequencing on the Illumina MiSeq platform for paired-end 151 bp sequencing. All sequencing was performed at the University of Massachusetts Boston using V2 chemistry.

**Sample preservation and extraction method verification.** Owing to the nature in which the samples were preserved (RNAlater and dry ice) and how the nucleic acids were extracted (MoBio PowerSoil DNA Isolation kit and following Mettel et al.[33]), we sought to understand the effect of our preservation and extraction methodology. While studies have shown RNAlater to be an effective method to preserve rRNA and mRNA for sequencing[35], these studies focused on water samples, and the efficacy of RNAlater in preserving sediment samples has not been investigated. To test our methodology, we extracted three salt marsh sediment samples preserved in either RNAlater or flash frozen in liquid nitrogen. In addition, we verified our extraction methods against a commercially available co-extraction kit (MoBio PowerSoil Total RNA Isolation Kit) following the manufacturer's instructions. The 16S rRNA gene and rRNA were sequenced and the resulting sequence data were processed as described below. To determine any significant differences between methods we compared weighted UniFrac similarity (Supplementary Fig. 3) values using a one-way ANOVA. There were no significant differences as a result of the different extraction or sediment preservation methods (Supplementary Fig. 3). It is also worth noting that the synchronous patterns in active microbial community structure in the reference marshes over a decade provide confidence that the long-term storage of sediments in RNALater did not adversely affect the nucleic acids (Fig. 1c).

**Sequence and data analysis.** Paired-end reads were joined using fastq-join[36] with default parameters. Joined reads were then demultiplexed and quality filtered in QIIME[37] following methods outlined previously[38]. Sequences were screened for chimeras using de novo mode in UCHIME[39] and the resulting chimeric sequences were discarded. After quality filtering, a total of 25.31 million rRNA and rRNA gene sequences were included in the final analysis. Operational taxonomic units (OTUs) were clustered at 97% sequence similarity using swarm[40] in QIIME, and OTUs appearing only once (singletons) across the data set were discarded. A representative sequence was chosen from the most abundant sequence for each out, and taxonomy was assigned using BLAST against the GreenGenes database (version 13.5). In addition, we filtered out all sequences matching chloroplasts. Representative sequences for each OTU were aligned with PyNast[41] and a phylogenetic tree was constructed using fasttree[42].

Beta diversity was calculated on normalized OTU tables using weighted UniFrac[43]. For comparison, we also analysed beta diversity using Bray Curtis similarity, which does not take the phylogenetic information of the OTUs into account when calculating similarity and the results are consistent with UniFrac results. To assess differences in community composition, we used adonis[18] with 10,000 permutations implemented in QIIME. Adonis tests significance of categorical variables using a PERANOVA by fitting linear models to distance matrices and assessing model fit with an F-test. To assess the degree of dormancy we calculated the 16S rRNA:16S rRNA gene +1 ratio for each taxa in each sample and defined a taxon as active with a ratio >1 (ref. 19). Work has shown 16S rRNA:16S rRNA gene ratios can vary among taxonomic groups[15] and the ratio used here, and previously[25], may produce many false positives for active taxa due to the variation in the ratio and biases associated with methodology[7,15]. To provide further support for our dormancy conclusions we addressed this issue by increasing the ratio of 16rRNA to the 16S rRNA gene between 1 and 50 to assess the effects of the ratio on the calculated rates of dormancy (Supplementary Fig. 2). We tested how levels of dormancy varied with fertilization using a one-way ANOVA in R (ref. 44).

Shannon diversity was calculated on rarified OTU tables normalized to the lowest sampling depth (32,501) in QIIME (10,000 restarts with steps of 100). To assess the diversity of active taxa, we calculated the 16S rRNA to 16S rRNA gene ratio for each order and defined an order as active when its ratio was >1. To determine significant changes in diversity between fertilization regimes and over

time we used multiple one-way ANOVAs after testing that the data met the relevant assumptions. Finally, to assess significant differences in relative OTU abundance and relative abundance of microbial orders due to fertilization we used a Kruskal–Wallis test in QIIME and R, defining significance with a Bonferroni corrected P value <0.001. Owing to the large number of OTUs present in the data set, we only compared taxa present 100 times when examining differential frequencies of taxa. Owing to the conservative nature of Bonferroni correction, we also examined the results of a false detection method (Benjamini–Hochberg test) and found that they produced identical results.

**Net ecosystem metabolism in tidal creeks.** YSI 6 series water property sondes (measuring conductivity, temperature and oxygen saturation state) and Onset HOBO pressure loggers (for water depth) were deployed in summer of 2012 in the bed of each tidal creek roughly 2 m below the tall-form S. alterniflora zone, at 115 m (N-enriched) and 129 m (reference) upstream from the nearest confluence with another primary creek. Sonde data were pre- and post-calibrated, and independently cross-calibrated against in situ oxygen concentration. Concentration is determined from saturation state using the oxygen solubility function[45]. Only 6 days of sensor data are analysed here because a bank collapse covered the sensor in the fertilized creek preventing further data analysis. Water depth and creek geometry were used to construct tidally varying water balances. Modelled freshwater inputs[46], tidal porewater exchange[47] and spring-neap-driven drainage of the marsh platform[48] are small compared with advective fluxes during the deployment period ($\sim 1\%$ of total flux each) and are not included.

Dissolved oxygen concentrations and the water budget are used to build a non-steady-state oxygen mass balance for each tidal creek[49]. The oxygen balance includes terms for gas exchange and the balance between photosynthesis and respiration, that is, net ecosystem metabolism in oxygen units (NEM), and is solved for a control volume at the sampling location. We extend the control surfaces into the marsh sediments to include any oxygen consumption by other terminal metabolic products (for example, sulfide[50,51]) in the resulting estimate of NEM. We chose a gas exchange parameterization that incorporates short-term variability in both wind speed and current velocity in a shallow, limited fetch channel[52]. Gas exchange flux is smaller in magnitude (25–45% lower) if the current and wind speed or current-only parameterizations are used[52,49]. All four estimates are consistent with in situ noble gas fluxes during the experimental period at each site.

Average and s.d.'s of NEM (per unit length of creek) over the hour of high tide at both creeks are plotted in Fig. 5a, when the tall-form S. alterniflora zone is flooded and interacting with the creek water column. Morning high tides may be more heterotrophic in this particular zone because of oxygen deficit built up overnight or light and temperature enhanced respiration before peak photosynthesis; this trend is consistent with the longer time series available at the reference creek. The difference between the NEM in the N-enriched and reference creeks is plotted in Fig. 5b. The generally higher daytime and lower night-time NEM in the N-enriched creek is consistent with both enhanced oxygen production and respiration in the N-enriched creek.

There are two potential sources of positive bias that may explain NEM values >0 for the three consecutive night-time spring tides between 23:00 and 01:00: first, we do not constrain bubble injection[53,54] and the dissolution of air trapped in the surface sediments as the tide rises; our mass balance would erroneously attribute oxygen added in this way to biological production. This process is likely to affect both creeks in a similar way and thus would not affect the NEM difference between the creeks (which is what is relevant to the arguments presented here). Second, the reference creek connects to an adjacent drainage during the highest tides (marked with an arrow on Fig. 5a,b) and therefore the volume transport model likely is not accurate at these highest tides. This factor would only affect the reference creek. On the basis of two additional weeks of $O_2$ measurements at the reference creek only (not shown), we can deduce from the difference of typical night-time high-tide NEM ($n=8$) and peak flooding NEM ($n=3$) that this second factor is likely equal to $\sim 0.35\,mmol\,O_2\,m^{-1}\,min^{-1}$. Thus, the errors caused with drainage during high tide would decrease the difference between reference and enriched creek but for two out of the three peak flooding time points (marked with black arrows), the night-time NEM difference between reference and enriched creeks would still be significant.

**Data availability.** The data associated with the net ecosystem metabolism measurements are available through the LTER Network Data Portal, PASTA (https://portal.lternet.edu/nis/home.jsp) with DOI: http://dx.doi.org/10.6073/pasta/fe47a9461bd332fae3ac7792af21c2b0, on the Plum Island Ecosystems LTER web site (http://pie-lter.ecosystems.mbl.edu). All microbial sequences have been deposited in the Sequence Read Archive under accession number SRP072677, and additional metadata can be accessed through the PIE LTER web site (http://pie-lter.ecosystems.mbl.edu).

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

## Acknowledgements

We thank researchers of the TIDE Project (NSF OCE0924287, OCE0923689, DEB0213767, DEB1354494 and DEB1353140) for maintaining the study site and the N-enrichment experiment. The TIDE project provided the YSI sondes, Jimmy Nelson did pre- and post-calibrations on the sondes, and Will Kearney and Sergio Fagherazzi provided the creek geometry transects for the water budget. We thank members of the Bowen lab for their help in field collections. Thom Thera and Illumina tech support provided tech support for sequencing. Bioinformatics could not have been done without the use of the supercomputing facilities managed by Jeff Dusenberry and the Research Computing Department at the University of Massachusetts Boston. The Plum Island LTER (NSF LTER OCE 0423565 and NSF OCE 1058747) provided critical intellectual and study site support. Finally, we thank Anne Giblin, David Johnson, James Nelson, Sarah Feinman, Andrew Babbin, Jay Lennon and an anonymous reviewer for helpful comments and discussions on the manuscript.

## Author contributions

L.A.D. designed and oversaw the nutrient enrichment experiment and J.L.B. designed the microbial component of the work; P.J.K., J.H.A. and J.L.B. collected samples; P.J.K. processed samples and performed bioinformatics analyses; E.M.H. and R.H.R.S. collected and analysed the oxygen data and provided estimates of ecosystem metabolism; P.J.K. and J.L.B. wrote the paper with contributions from L.A.D., E.M.H. and R.H.R.S.

## Additional information

**Competing financial interests:** The authors declare no competing financial interests.

