## [Peer Review File · Nature Communications]

Reviewer Comments:

Reviewer #1 (Remarks to the Author):

Overview:

The manuscript by Kearns at all examines the response of saltmarsh bacterial communities to long-term experimental nitrogen (N) additions. The authors use 16S rRNA genes (i.e., "DNA") and 16S rRNA transcripts (i.e., "RNA") to characterize compositional changes in the bacterial communities to N addition, but also the effect of time (year since fertilization began) and habitat (low vs. high marsh). The authors report that the proportion of dormant taxa increases in response fertilization and the diversity of this dormant pool declines. A number of studies over the last few years have documented that the composition of DNA- and RNA-based microbial communities are different suggesting that many bacterial taxa may be in a low or inactive state and thus not contributing to ecosystem processes. In this interesting study, Kearns et al. report that global change drivers like nutrient enrichment may regulate the metabolic activity of key microbial populations and functional groups in ways that may have effects on the biogeochemistry of important ecosystems.

Major comments:

The most compelling results reported in the Kearns et al. ms might be that 1) they find the "highest proportion of dormancy yet reported", and 2) that dormancy is significantly higher in the enriched vs. reference sites. Since I have wrestled with similar questions, I'm also aware of some of the criticisms that will likely come up when this paper is published. So, I'll raise some of these here in hopes that they will make for a stronger manuscript. I'll start with methods:

- It sounds like some of the samples may have been stored in RNAlater for a long time, perhaps a decade. Is this OK? I suppose it shouldn't account for the DNA vs. RNA compositional differences.
- Could the RNA extraction and processing described on 301-322 contribute to the differences in composition that is reported for the DNA and RNA compositional differences?

One could also ask about the calculation that was used to estimate dormancy. In the paper, Kearns et al. classify a taxon as dormant if the 16S rRNA:16S rRNA gene ratio is >1 . How sensitive is the dormancy metric to this assumption? I have a former postdoc who used similar procedures (ms in prep; happy to share). I asked him to "titrate" the 1:1 ratio and see if the findings were robust; obviously the magnitude of the response (proportion dormant) changed, but the effect size did not, which was reassuring. The authors discuss the alternate method, which is intended to bolster the principal findings. I didn't completely follow this in the main text (lines 98-100), but it became clearer in the Methods. Perhaps this could be addressed in a revision. Specifically, it wasn't entirely obvious how this approach addressed the caveats that are mentioned on line 100. Last, in Figure 3 x-axis, there's a "+1", I'm guessing this is meant to deal with taxa for which there is RNA recovery but not DNA, so you are assuming it's there but below detection?

I also have a question about the ecological interpretation of the major result. Dormancy is something that arises when individuals are energetically or nutritionally limited, which is in line with the authors general statements. Why is it then, that in salt marsh ecosystems -- which I presume are N-limited -- would bacteria become more dormant with more N? This seems to be the opposite of what one might expect. In general, the paper would be stronger if there was a more mechanistic framework to help explain their major finding. There is an attempt to address this on lines 134 - 145 based on shifts in autotrophy and heterotrophy. The authors seem to want to link this pattern to shifts in the active community. Although this model sounds somewhat reasonable, I'm not sure it's strongly supported by the data. The methods section seems to spend considerable space describing the sondes and model used to estimate metabolism. Certainly, this is a cool approach, but I wasn't entirely convinced that it was essential for the main thesis.

Somewhat related, the authors found that Cyanobacteria and Oscillatoria were disproportionately active in the fertilized treatment. This struck me as a bit strange, too, because at least some of these bacteria are capable of nitrogen fixation, a costly strategy that one would not predict if the environment were replete with N. Perhaps there are other advantages to being a cyanobacterium in an N-rich environment and other studies which support this trend.

I had some comments about the figures. Starting on line 89, the authors state "Analysis of the active bacterial community indicates that long-term nutrient enrichment spatiotemporally standardizes the active community, overriding the importance of both habitat and season as structuring forces" However, in Figure 1C, there seems to be discernable structure within the "All Fert" ellipse. How were these ellipses generated? Are they consistent with the PERMANOVA results? Also, abbreviations for treatments are not provided in Figure 1 (e.g. TSA, SP).

Other figure comments: I did not find the "lead-in" sentences of caption 3 or 4 to be that inviting to a general reader. Figure 2 might be clearer if a single format was chosen for both panels. Also on Figure 2, the mix of regression and ANOVA described in caption is sort of odd. If regression is used (appropriate with year data and gaps) perhaps consider an indicator or dummy variable that treats time as continuous Fert vs. ref as categorical variables. However, habitat also appears to be in the mix. How is this being dealt with given its importance in Fig 1? In figure 4, the figure seems to show that there are taxa that are abundant active and abundant dormant, but it is not clear how this figure was put together. What were the criteria? In the abstract and last paragraph of the paper, there is a comparison to effects of fertilization in salt marshes to grasslands. I'm not really sure I understood this. Given the location of these statements, my sense is that this is supposed to be important. Perhaps a revision of this would help clarify the intended point.

Other thoughts:

- Abstract: "Shifts in the active taxa we observe are surprising because the total microbial community in marsh sediments was unaffected by nitrogen enrichment, even after a decade of N supply" -- what does this say about the size and longevity of microbial biomass pool?
- Line 119 and elsewhere. "Abundance" should probably be "relative abundance". And by "abundance" do the authors mean DNA?
- Line 120 -122. Discussion on rarity and commonness is interesting; perhaps elaborate?
- Line 129: "which may allow better adaptation to environmental perturbations" somewhat awkward in an evolutionary sense.
- Line 253 and elsewhere. Bonferroni is notoriously conservative. Should look into other (easy) false discovery rate corrections like Benjamini-Hochberg
- Line 257 (and figure) natural log? log 10? log 2?
- Line 349: regarding discarding of singletons, was this done for the entire data set or on a per sample basis?
- Line 356: "taxonomic information of the OTU" do you mean "phylogenetic"?

Jay T. Lennon, Indiana University

Reviewer #2 (Remarks to the Author):

Kearns et al. present an interesting story regarding the influence of N enrichments in salt marshes. Specifically they examine the effect on bacterial communities - both total and active. What makes this study unique is that they are able to assess the percent dormant of the community- between ambient and fertilized treatments- as many studies only focus on the total communities. Overall, the manuscript is well written, concise and clear. However, in the current state it is lacking a component that would make it more accessible to a broader audience. The data and methods are sound and appropriate, as are the statistics. Conclusions, while sound, are lacking a connection to a broader scope. This could be addressed by either comparing/relating salt marshes to other ecosystems. Or addressing more clearly why N deposition in this system is so important. This is still only one salt marsh and a more global viewpoint would make the conclusions more applicable. Further, in the current draft it is not quite clear how changes to the bacterial community will affect ecosystem functions- though this is stated in the opening paragraph. Further clarification, or reassessment of the

functional data is needed.

Specific comments:

Line 19: This idea about biodiversity and ecosystem function is very relevant but the research does little to address these topics. To make this paper more applicable I would encourage the authors to add a stronger component linking biodiversity and ecosystem function. Possibly there is a way to tie these specific results into a broader global context.

Line 25-27: Explain in the main text how a decrease in biodiversity and increase in dormancy could affect ecosystem functions

Line 28-29: In the main text could there be more explanation on why the total community stays consistent over 10 years, but the active community shifts. At some point wouldn't you expect to see this change in the total community? If not, why does it persist?

Line 36-37: This is a compelling point that could use more focus in the main text.

Line 40: put this in a more global/larger context. How much area? Or how much Nitrogen relative to other ecosystems. Add a sentence to demonstrate why are salt marshes so important.

Line 43-45: It is not clear why an increase in respiration would be directly related to community shifts? Why not just a more active, but same community. A stronger link is needed to explain why this study was done, and what makes it novel.

Line 51-57: Still, it is unclear why this would be related to increased respiration rates mentioned above.

Line 59: an explanation is needed about the importance of dormant bacteria, and how a high or low proportion would influence ecosystem function/biodiversity/etc.

Line 61: Possibly change this sentence to better explain DNA vs RNA = total vs active (as done farther down in the methods). Then this section is more accessible by those less familiar with these techniques.

Line 81: Start with the result first, then explain the figure (PCA plot) used to show the result.

Line 87-88: would be nice to do a bit of speculation on why this is and what this means for bacterial communities/salt marshes.

Line 89-93: This is an interesting concept that could be expanded on more in the introduction as part of the hypotheses.

L103-107: Expand on why this is important.

L108-113: Similarly, what are the ecosystem consequences of this increased dormancy/decreased activity

L134-145: this section could be set up more in the introduction to really highlight the ecosystem implications of the changes observed. Play it up.

L152: why another comparison to grasslands? As previously explained these are drastically different systems. Instead maybe results could be likened to marine or lake systems, to extend the impact found here?

Fig 1: in figure legend refer to as DNA and RNA - like in the actual figure.

Fig 4: This is a great plot- however, as with the others, is there another color choice that can be used? Red/blue is harsh.

Ext data Fig 3: the PDF text is jumbled

Reviewer #1 (Remarks to the Author):

Overview:

The manuscript by Kearns et al. examines the response of saltmarsh bacterial communities to long-term experimental nitrogen (N) additions. The authors use 16S rRNA genes (i.e., "DNA") and 16S rRNA transcripts (i.e., "RNA") to characterize compositional changes in the bacterial communities to N addition, but also the effect of time (year since fertilization began) and habitat (low vs. high marsh). The authors report that the proportion of dormant taxa increases in response to fertilization and the diversity of this dormant pool declines. A number of studies over the last few years have documented that the composition of DNA- and RNA-based microbial communities are different suggesting that many bacterial taxa may be in a low or inactive state and thus not contributing to ecosystem processes. In this interesting study, Kearns et al. report that global change drivers like nutrient enrichment may regulate the metabolic activity of key microbial populations and functional groups in ways that may have effects on the biogeochemistry of important ecosystems.

We thank Dr. Lennon for his extremely helpful review, which greatly enhanced the quality of this manuscript.

Major comments:

The most compelling results reported in the Kearns et al. ms might be that 1) they find the "highest proportion of dormancy yet reported", and 2) that dormancy is significantly higher in the enriched vs. reference sites. Since I have wrestled with similar questions, I'm also aware of some of the criticisms that will likely come up when this paper is published. So, I'll raise some of these here in hopes that they will make for a stronger manuscript. I'll start with methods:

- It sounds like some of the samples may have been stored in RNAlater for a long time, perhaps a decade. Is this OK? I suppose it shouldn't account for the DNA vs. RNA compositional differences.

We also had concerns about this method (see paragraph below), however RNAlater has been shown to produce robust RNA profiles, See: Ottesen et al (2011) The ISME Journal 5, 1881–1895

- Could the RNA extraction and processing described on 301-322 contribute to the differences in composition that is reported for the DNA and RNA compositional differences?

We had a similar set of concerns regarding the extraction method as well as the samples being stored in RNAlater. To address this, we compared our extraction protocols to a commercially available DNA/RNA co-extraction kit (MoBio PowerSoil Total RNA Isolation Kit) as well as RNAlater treated samples to samples flash frozen in liquid nitrogen. Our results comparing weighted UniFrac similarities suggest that our results are comparable to a co-extraction kit and that samples treated with RNAlater displayed indistinguishable patterns from samples treated with liquid nitrogen (see the new SI figure 3). While we did not explicitly test the long-term effects, the robustness of the results we present in Fig. 1C suggest that the long-term effects are negligible. The active community in the reference marshes (blue points in Fig. 1C) show repeatable clustering patterns over both season and time (eg. May-June samples from 2005 and 2006 cluster with May-June samples from 2013 and 2014), that would be unlikely if the RNA were degraded over time. We have made these points in the manuscript on lines 116-118 and 298-308.

One could also ask about the calculation that was used to estimate dormancy. In the paper, Kearns et al.

classify a taxon as dormant if the 16S rRNA:16S rRNA gene ratio is >1. How sensitive is the dormancy metric to this assumption? I have a former postdoc who used similar procedures (ms in prep; happy to share). I asked him to "titrate" the 1:1 ratio and see if the findings were robust; obviously the magnitude of the response (proportion dormant) changed, but the effect size did not, which was reassuring. The authors discuss the alternate method, which is intended to bolster the principal findings. I didn't completely follow this in the main text (lines 98-100), but it became clearer in the Methods. Perhaps this could be addressed in a revision. Specifically, it wasn't entirely obvious how this approach addressed the caveats that are mentioned on line 100.

We have removed this analysis in favor of the method suggested by Dr. Lennon as we believe it provides a more robust method for analyzing dormancy. We have 'titrated' our DNA/RNA ratio increasing the ratio used to define active from 1 all the way up to 50. We have created a new figure for the supplemental (SI figure 2) and have added text to the main text (Lines 124-129) and methods (lines 330-333). Our results suggest that regardless of the ratio used, our results are robust. We agree with the reviewer that this is a much more robust way to test assumptions about the definition of dormancy and have replaced our previous supplemental figure 2 with this iteration.

Last, in Figure 3 x-axis, there's a "+1", I'm guessing this is meant to deal with taxa for which there is RNA recovery but not DNA, so you are assuming it's there but below detection?

Yes, this is correct. We've added a line in the text (lines 141-143) and methods (line 327) to address this.

I also have a question about the ecological interpretation of the major result. Dormancy is something that arises when individuals are energetically or nutritionally limited, which is in line with the authors general statements. Why is it then, that in salt marsh ecosystems -- which I presume are N-limited -- would bacteria become more dormant with more N? This seems to be the opposite of what one might expect. In general, the paper would be stronger if there was a more mechanistic framework to help explain their major finding. There is an attempt to address this on lines 134 - 145 based on shifts in autotrophy and heterotrophy. The authors seem to want to link this pattern to shifts in the active community. Although this model sounds somewhat reasonable, I'm not sure it's strongly supported by the data. The methods section seems to spend considerable space describing the sondes and model used to estimate metabolism. Certainly, this is a cool approach, but I wasn't entirely convinced that it was essential for the main thesis.

We have revised this section, adding more detail about dormancy and why we see a differential response to nutrient enrichment than has been previously reported. Further, we have more thoroughly incorporated the measurements of NEM to our results. The revised text can be found on lines 189-198

Somewhat related, the authors found that Cyanobacteria and Oscillatoria were disproportionately active in the fertilized treatment. This struck me as a bit strange, too, because at least some of these bacteria are capable of nitrogen fixation, a costly strategy that one would not predict if the environment were replete with N. Perhaps there are other advantages to being a cyanobacterium in an N-rich environment and other studies which support this trend.

We have added discussion about this idea on lines 165-171 and 189-198 better linking the response of the bacteria to N-enrichment and discussing the physiology of Oscillatoria.

I had some comments about the figures. Starting on line 89, the authors state "Analysis of the active

bacterial community indicates that long-term nutrient enrichment spatiotemporally standardizes the active community, overriding the importance of both habitat and season as structuring forces" However, in Figure 1C, there seems to be discernable structure within the "All Fert" ellipse. How were these ellipses generated? Are they consistent with the PERMANOVA results? Also, abbreviations for treatments are not provided in Figure 1 (e.g. TSA, SP).

The ellipsis were not generated by any mathematical means, they were added to the figure to draw attention to the overarching patterns in the data. The ellipsis are, however, consistent with all PERMANOVA results discussed in the text and the figure legend. Additionally, we've added additional text to address the structure within the "All fert" ellipse on lines 118-122.

Other figure comments: I did not find the "lead-in" sentences of caption 3 or 4 to be that inviting to a general reader.

We have re-worded the figure legend leading sentences to make the legends more accessible to a wider audience.

Figure 2 might be clearer if a single format was chose for both panels. Also on Figure 2, the mix of regression and ANOVA described in caption is sort of odd. If regression is used (appropriate with year data and gaps) perhaps consider an indicator or dummy variable that treats time as continuous Fert vs. ref as categorical variables. However, habitat also appears to be in the mix. How is this being dealt with given its importance in Fig 1?

We re-created figure 2 to better incorporate the distinction between the grass types and make Figure 2B more in-line with Figure 2A (box and whisker plots). Additionally, we removed the linear regression in favor of a one-way ANOVA to deal with the issues reviewer one brings up. We've amended the text on lines 133-139 and in the methods on lines 337-338 to reflect this change.

In figure 4, the figure seems to show that there are taxa that are abundant active and abundant dormant, but it is not clear how this figure was put together. What were the criteria?

This was generated with a non-parametric ANOVA (Kruskal-Wallis test) and the criteria are listed in the figure legend as "(Kruskal-Wallis test, Bonferoni corrected $p < 0.0001$) present at least 100 times", we've added lines to the main text (lines 154-155), and the methods on lines 340-342.

In the abstract and last paragraph of the paper, there is a comparison to effects of fertilization in salt marshes to grasslands. I'm not really sure I understood this. Given the location of these statements, my sense is that this is supposed to be important. Perhaps a revision of this would help clarify the intended point.

This aspect was removed from the abstract. However, we've addressed this comment with a paragraph in the discussion on lines 202-212.

Other thoughts:

- Abstract: "Shifts in the active taxa we observe are surprising because the total microbial community in marsh sediments was unaffected by nitrogen enrichment, even after a decade of N supply" -- what does this say about the size and longevity of microbial biomass pool?

We've removed this text from the abstract, however we've addressed this comment on lines 108-111 with discussion about the stability of the community.

- Line 119 and elsewhere. "Abundance" should probably be "relative abundance". And by "abundance" do the authors mean DNA?

We have amended the manuscript to read 'relative abundance' in lieu of true abundance.

- Line 120 -122. Discussion on rarity and commonness is interesting; perhaps elaborate?

We have added discussion about this on lines 150-153.

- Line 129: "which may allow better adaptation to environmental perturbations" somewhat awkward in an evolutionary sense.

We have re-worded the sentence to read "which may allow faster response to environmental perturbations" on line 161.

- Line 253 and elsewhere. Bonferroni is notoriously conservative. Should look into other (easy) false discovery rate corrections like Benjamini-Hochberg

In addition to a Bonferroni p-value, we had also examined the p-value derived from a Benjamini-Hochberg test. For the results we present in the paper (Figure 3 and 4) the results are identical between the two methods as we look at the most abundant taxa present in the dataset. We've added text to the methods on lines 340-342.

- Line 257 (and figure) natural log? log 10? log 2?

Log 10, we have added this to the figure legend for clarity.

- Line 349: regarding discarding of singletons, was this done for the entire data set or on a per sample basis?

This was done on the whole dataset. So any OTU appearing once across the dataset was discarded from the dataset. We've amended the methods to reflect this on lines 314-315.

- Line 356: "taxonomic information of the OTU" do you mean "phylogenetic"?

Yes, however, this sentence was removed from the manuscript.

Jay T. Lennon, Indiana University

Reviewer #2 (Remarks to the Author):

Kearns et al. present an interesting story regarding the influence of N enrichments in salt marshes. Specifically they examine the effect on bacterial communities - both total and active. What makes this study unique is that they are able to assess the percent dormant of the community- between ambient and fertilized treatments- as many studies only focus on the total communities. Overall, the manuscript is well written, concise and clear. However, in the current state it is lacking a component that would make it more accessible to a broader audience. The data and methods are sound and appropriate, as are the statistics. Conclusions, while sound, are lacking a connection to a broader scope. This could be addressed

by either comparing/relating salt marshes to other ecosystems. Or addressing more clearly why N deposition in this system is so important. This is still only one salt marsh and a more global viewpoint would make the conclusions more applicable. Further, in the current draft it is not quite clear how changes to the bacterial community will affect ecosystem functions- though this is stated in the opening paragraph. Further clarification, or reassessment of the functional data is needed.

We thank the reviewer for his/her helpful comments on the manuscript. Throughout the manuscript, and in particular on 44-47, 70-73, and 213-224 we have tried to expand the global scope of our study by 1) providing context for the nutrient enrichment status of marshes globally on lines 52-56 and by expanding or discussion of the functional implications of our work on lines 215-217. Responses to specific queries are below.

Specific comments:

Line 19: This idea about biodiversity and ecosystem function is very relevant but the research does little to address these topics. To make this paper more applicable I would encourage the authors to add a stronger component linking biodiversity and ecosystem function. Possibly there is a way to tie these specific results into a broader global context.

We have incorporated more biodiversity/ecosystem function throughout the paper, in particular on lines 44-47, 70-73, and 213-224, to better place our research in a larger, global context

Line 25-27: Explain in the main text how a decrease in biodiversity and increase in dormancy could affect ecosystem functions

We have added this discussion on lines 70-73 and 215-224.

Line 28-29: In the main text could there be more explanation on why the total community stays consistent over 10 years, but the active community shifts. At some point wouldn't you expect to see this change in the total community? If not, why does it persist?

Given the body of evidence on the effects of nutrient enrichment on microbial community composition, our results are surprising. We do not have a concrete answer for the reviewer, however, work in another New England salt Marsh has demonstrated that even after 40 years of fertilization there was no net change in the total microbial community (Bowen et al. 2011; ISME J). Suggesting marsh communities are stable and resistant to perturbations on decadal time frames. We've added text to the manuscript on lines 108-111.

Line 36-37: This is a compelling point that could use more focus in the main text.

We've added discussion about this point on lines 196-202 and 209-212.

Line 40: put this in a more global/larger context. How much area? Or how much Nitrogen relative to other ecosystems. Add a sentence to demonstrate why are salt marshes so important.

We have added further discussion about this point to the introduction on lines 52-56.

Line43-45: It is not clear why an increase in respiration would be directly related to community shifts? Why not just a more active, but same community. A stronger link is needed to explain why this study was done, and what makes it novel.

We have expanded on this on lines 56-58 and 60-62 to provide further clarity on the matter.

Line 51-57: Still, it is unclear why this would be related to increased respiration rates mentioned above.

We have expanded on this on lines 60-62 to provide further clarity on the matter.

Line 59: an explanation is needed about the importance of dormant bacteria, and how a high or low proportion would influence ecosystem function/biodiversity/etc.

We have added details to the text on lines 70-73.

Line 61: Possibly change this sentence to better explain DNA vs RNA = total vs active (as done farther down in the methods). Then this section is more accessible by those less familiar with these techniques.

We have revised this section to better explain the rationale for sequencing both products on lines 82-83.

Line 81: Start with the result first, then explain the figure (PCA plot) used to show the result.

We have reworked this sentence to lead with the result, as the reviewer suggests on lines 102-104.

Line 87-88: would be nice to do a bit of speculation on why this is and what this means for bacterial communities/salt marshes.

We have expanded upon this point as the reviewer suggest on lines 108-111.

Line 89-93: This is an interesting concept that could be expanded on more in the introduction as part of the hypotheses.

We have added this idea to our hypotheses on lines 79-81.

L103-107: Expand on why this is important.

We have expanded on this idea on lines 215-224 in the discussion.

L108-113: Similarly, what are the ecosystem consequences of this increased dormancy/decreased activity

We have added a paragraph discussing the consequences of our results on lines 70-73 in the introduction and 215-224 in the discussion.

L134-145: this section could be set up more in the introduction to really highlight the ecosystem implications of the changes observed. Play it up.

We have expended upon this in the introduction on lines 161-168 and 177-182.

L152: why another comparison to grasslands? As previously explained these are drastically different systems. Instead maybe results could be likened to marine or lake systems, to extend the impact found here?

This was removed from the abstract due to space constrictions, however we've addressed this comment in the discussion on lines 202-212.

Fig 1: in figure legend refer to as DNA and RNA - like in the actual figure.

To keep the figure consistent with the rest of the text, where we use 16S rRNA gene and 16S rRNA, we have modified the figure legend of 1 to read '16S rRNA and 16S rRNA gene'.

Fig 4: This is a great plot- however, as with the others, is there another color choice that can be used? Red/blue is harsh.

We agree the chart is difficult on the eyes. As with the previous plots we have changed the red to a darker color and for easier readability we have increased the space between the bars.

Ext data Fig 3: the PDF text is jumbled

In the newest submitted version, we have ensured this figure is present in the intended condition.

REVIEWERS' COMMENTS:

Reviewer #1 (Remarks to the Author):

Overview: Kearns et al. have done a good job of revising their manuscript. However, I have a few remaining and additional comments that will hopefully be useful.

Major points:

-- Lines 76-83. There is some discussion about how marshes are naturally dynamic systems. Temporal fluctuations in environmental drivers may be important for favoring dormancy as a life history strategy as theory suggests (e.g. Malik and Smith 2008 and other general ecological work). Kearns et al. argue that nutrient enrichment "overwhelms patterns of activity" and presumably the importance of natural variability found in reference marshes. It seems to me that this would lead to the prediction that because enriched marshes are less temporally variable, that dormancy would be less prevalent.

-- Similarly, on lines 196-200, I wasn't entirely convinced by the explanation provide for why dormancy would be higher in the N-enriched marshes from a resource-based perspective. Kearns et al. invoke competition and bloom dynamics (which is somewhat speculative), in a way that wasn't entirely clear to me.

-- In sum, the patterns in the paper are very strong, but I'm left a little less than 100% satisfied by the ecological interpretation as to why N-limited marshes become dominated by dormancy when N is added. Somewhat of a conundrum. Perhaps there are other non-resource based explanations for this pattern. After decades of fertilization, do the marshes change in some other way (e.g., physically) that might affect microbial activity?

-- Line 129: I'm glad that Kearns et al. found my suggestion for using different rRNA : rDNA ratios useful for testing the robustness of their findings. It would be appropriate to cite Aanderud et al. (2016) where this approach was first used: <http://journal.frontiersin.org/article/10.3389/fmicb.2016.00853/full>

Minor points:

-- Lines 125: Delete double period at end of sentence

-- Line 53-156: Check grammar and/or polish for clarity

-- Line 159-161: More appropriate to say that retrieved sequences are closely related to organisms with said metabolisms.

-- Line 182: Somewhat confusing to talk about Cyanobacterial dominance in N-enriched cites and call this "algal productivity". Primary productivity? Phototrophic productivity?

Jay T. Lennon

Reviewer #2 (Remarks to the Author):

I am re-reviewing this manuscript. Briefly, the study explores the effect of N enrichment on salt marsh bacterial communities. What makes this study unique is that the authors look at both the active and total microbial community, and present the results into a broader ecological context regarding the role of bacterial community diversity and activity in ecosystem processes. In my previous comments I encouraged the authors to place the results into a broader context. Overall I think they have done this very well, and I really enjoyed reading this new version. However, while the manuscript now fits more with Nature Comm. it can still be strengthened in a few key places to focus on the novel components of the study. Specifically, the fertilization and salt marsh components of the study, while interesting, are not the novel components. Rather the salt marsh seems to be a good system to study bacterial diversity/activity/dormancy in relation to critical ecosystem processes. Further, these results give important insight into microbial communities, and that knowledge can be used in other systems. I have provided more specific comments below.

The abstract could use a bit more focus- setting up the bacterial dormancy story rather than just anthropogenic nitrogen inputs. One idea, would be to start with line 21, and then introduce the idea of active/dormancy in a second sentence and how so much is not known. This will also give better context to the last few sentences and give a more powerful message. There are likely a few other ways to approach this- but I think this is the novel part of the study.

L65-75: Paragraph 3 should be moved up to paragraph 2. Currently the paragraph about salt marshes - while relevant - does not have a smooth transition. Then this would set up the hypothesis in paragraph 4 more clearly. Alternatively, line 51-53 could be rewritten to flow into the second paragraph.

L73-75: This is a very strong and important statement, and this idea should be reiterated throughout the manuscript.

L125: delete "."

L178: Fig 5 - This figure doesn't particularly strengthen the main findings and could be put in the supplement.

L184-185: "portion of the carbon cycle" is a bit speculative and wishy washy. Can this be restated to clarify?

L201: how does this relate back to marsh collapse?

L223-224: Has this been seen elsewhere? And is the increase in autotrophy/heterotrophy detectable at a large ecosystem scale? By addressing these two questions this idea about microbial diversity and ecosystem function could be made stronger.

L230: Are dormancy values between studies comparable? Are there not methodological caveats that prevent this? For example, with the rRNA gene comparing the relative abundance the different taxa isn't

usually appropriate as there are so many methodological differences between studies that create differences in abundance (See Iozupone et al. 2014).

Fig 2: add 'Year' on x-axis

Both Fig 3 and Fig 4 are not doing much to strengthen the story. They could both be simplified to highlight the key/most interesting taxa and the full figures can go in the supplement. As presented it is simply too much information to quickly see the main points.

Fig 3: should Chromatiales be colored black??

Fig 4: This isn't quite clear. Maybe only focus on the 5-10 most important OTUs.

Reviewer #1 (Remarks to the Author):

Major points:

-- Lines 76-83. There is some discussion about how marshes are naturally dynamic systems. Temporal fluctuations in environmental drivers may be important for favoring dormancy as a life history strategy as theory suggests (e.g. Malik and Smith 2008 and other general ecological work). Kearns et al. argue that nutrient enrichment "overwhelms patterns of activity" and presumably the importance of natural variability found in reference marshes. It seems to me that this would lead to the prediction that because enriched marshes are less temporally variable, that dormancy would be less prevalent.

We have reworded this sentence to read "We hypothesize that excess N will favor a small number of taxa that are able to respond to increased N availability and ultimately result in an increase in the portion of dormant cells."

-- Similarly, on lines 196-200, I wasn't entirely convinced by the explanation provide for why dormancy would be higher in the N-enriched marshes from a resource-based perspective. Kearns et al. invoke competition and bloom dynamics (which is somewhat speculative), in a way that wasn't entirely clear to me.

-- In sum, the patterns in the paper are very strong, but I'm left a little less than 100% satisfied by the ecological interpretation as to why N-limited marshes become dominated by dormancy when N is added. Somewhat of a conundrum. Perhaps there are other non-resource based explanations for this pattern. After decades of fertilization, do the marshes change in some other way (e.g., physically) that might affect microbial activity?

We have expanded upon these two points as suggested by Dr. Lennon on lines 210-212 to try and further clarify the concern he raised. We also have added some text discussing some of the other changes occurring in the marsh due to fertilization on lines 224-226.

-- Line 129: I'm glad that Kearns et al. found my suggestion for using different rRNA : rDNA ratios useful for testing the robustness of their findings. It would be appropriate to cite Aanderud et al. (2016) where this approach was first used:
<http://journal.frontiersin.org/article/10.3389/fmicb.2016.00853/full>

We have added the citation to the main text as well as the methods.

Minor points:

-- Lines 125: Delete double period at end of sentence
We have removed the second period.

-- Line 53-156: Check grammar and/or polish for clarity
We have edited this section to provide more clarity.

-- Line 159-161: More appropriate to say that retrieved sequences are closely related to organisms with said metabolisms.
We have reworded the sentence to read "N-enriched sediments contained large numbers of

sequences that were closely related to anaerobic sulfate reducers from the Deltaproteobacterial order Desulfobacterales as well as to numerous autotrophic Cyanobacteria from the order Oscillatoriales”

-- Line 182: Somewhat confusing to talk about Cyanobacterial dominance in N-enriched sites and call this "algal productivity". Primary productivity? Phototrophic productivity?

We have changed this to read ‘phototrophic productivity’ as suggested by the reviewer.

Jay T. Lennon

Reviewer #2 (Remarks to the Author):

I am re-reviewing this manuscript. Briefly, the study explores the effect of N enrichment on salt marsh bacterial communities. What makes this study unique is that the authors look at both the active and total microbial community, and present the results into a broader ecological context regarding the role of bacterial community diversity and activity in ecosystem processes. In my previous comments I encouraged the authors to place the results into a broader context. Overall I think they have done this very well, and I really enjoyed reading this new version. However, while the manuscript now fits more with Nature Comm. it can still be strengthened in a few key places to focus on the novel components of the study. Specifically, the fertilization and salt marsh components of the study, while interesting, are not the novel components. Rather the salt marsh seems to be a good system to study bacterial diversity/activity/dormancy in relation to critical ecosystem processes. Further, these results give important insight into microbial communities, and that knowledge can be used in other systems. I have provided more specific comments below.

The abstract could use a bit more focus- setting up the bacterial dormancy story rather than just anthropogenic nitrogen inputs. One idea, would be to start with line 21, and then introduce the idea of active/dormancy in a second sentence and how so much is not known. This will also give better context to the last few sentences and give a more powerful message. There are likely a few other ways to approach this- but I think this is the novel part of the study.

We have updated the abstract to better incorporate dormancy as a central idea.

L65-75: Paragraph 3 should be moved up to paragraph 2. Currently the paragraph about salt marshes - while relevant - does not have a smooth transition. Then this would set up the hypothesis in paragraph 4 more clearly. Alternatively, line 51-53 could be rewritten to flow into the second paragraph.

We have moved the third paragraph to be in front of the second to provide smoother transitions as suggested by the reviewer.

L73-75: This is a very strong and important statement, and this idea should be reiterated throughout the manuscript.

We have lines addressing this comment on lines 201-204 and 234-238.

L125: delete "."

We deleted the extra period.

L178: Fig 5 - This figure doesn't particularly strengthen the main findings and could be put in the supplement.

We believe to better link the changes in dormancy to changes in ecosystem function (as per the request of the reviewer) this figure should remain in the main text.

L184-185: "portion of the carbon cycle" is a bit speculative and wishy washy. Can this be restated to clarify?

We have clarified this sentence to read "Thus, there may be a shift in a portion of the carbon cycle in nutrient enriched marshes due to enhancement of both bacterial photosynthesis and respiration."

L201: how does this relate back to marsh collapse?

We have added lines addressing this comment on lines 208-209.

L223-224: Has this been seen elsewhere? And is the increase in autotrophy/heterotrophy detectable at a large ecosystem scale? By addressing these two questions this idea about microbial diversity and ecosystem function could be made stronger.

No this has not been seen elsewhere. Our results (Fig. 5) do seem to indicate that there are changes in function. We also believe this is why figure 5 should remain in the main text.

L230: Are dormancy values between studies comparable? Are there not methodological caveats that prevent this? For example, with the rRNA gene comparing the relative abundance the different taxa isn't usually appropriate as there are so many methodological differences between studies that create differences in abundance (See Iozupone et al. 2014).

While this is true and that methodology can induce biases, we limited the scope of our comparison to only those using molecular-based technology where the biases should be less of a hindrance to the comparisons.

Fig 2: add 'Year' on x-axis

We added year to both x-axes.

Both Fig 3 and Fig 4 are not doing much to strengthen the story. They could both be simplified to highlight the key/most interesting taxa and the full figures can go in the supplement. As presented it is simply too much information to quickly see the main points.

Although the level of detail presented in these figures may seem too much to people who are more broadly interested in the big picture (such as this reviewer), we also respectfully suggest that there may be others who are more interested in the details of the nature of the changes among the different microbial taxa. Thus, we would prefer to keep these figures present in the manuscript as is, unless the editorial team agrees they should be removed.

Fig 3: should Chromatiales be colored black??

Yes, we have fixed this error.

Fig 4: This isn't quite clear. Maybe only focus on the 5-10 most important OTUs.

See comments above.